# Lempel-Ziv Penalty: An information-theoretic repetition penalty for autoregressive language models

**Antonio A. Ginart, Naveen Kodali, Jason Lee, Caiming Xiong, Silvio Savarese, John Emmons**
**Salesforce AI Research**
`aginart@salesforce.com`, `jemmons@salesforce.com`

**Reviewed on OpenReview:** `https://openreview.net/forum?id=vNzPB4YCHj`

## Abstract

We introduce the Lempel-Ziv (LZ) penalty, a penalty specialized for reducing degenerate repetitions in autoregressive language models without loss of capability. The penalty is based on the codelengths in the LZ77 universal lossless compression algorithm. Through the lens of the prediction-compression duality, decoding with the LZ penalty has the interpretation of sampling from the residual distribution after removing the information that is highly compressible. We demonstrate that the LZ penalty enables open-source reasoning models to operate with greedy decoding without loss of capability and without instances of degenerate repetition. In contrast, the industry-standard frequency penalty and repetition penalty are ineffective, incurring degenerate repetition rates of up to 4% or more.

## 1 Introduction

There has been an advent in reasoning models (Singh et al., 2025; Yang et al., 2025; Guo et al., 2025). Reasoning models are a class of large, autoregressive foundation models that achieve impressive capability gains in certain domains by scaling chain-of-thought reasoning sequences at inference time. While reasoning models are a promising approach for scaling inference-time compute, open-source reasoning models currently suffer from some friction points that make their use problematic for downstream application developers due to a lack of determinism around the reasoning traces. This lack of determinism is rooted in the fact that reasoning models do not run well at low temperatures because the sampling distribution can mode collapse into degenerate repetitions.

Enabling deterministic algorithms for generation is useful for debugging and may be an explicit requirement for some deployments. Furthermore, even at higher temperatures, even frontier models can still fall into degenerate repetitions in real-world deployments, such as within Cursor[1].

There are two industry-standard penalties aimed at reducing repetition. First, the repetition penalty (Keskar et al., 2019) applies a fixed logit penalty that encourages the model to use new tokens. The frequency penalty is more subtle, and applies a logit penalty proportional to the token count in context. Neither penalty consistently stops degenerate repetitions without degrading the sample quality. First, the repetition penalty does not actually succeed in preventing degenerate repetitions because it applies a naive, binary modal penalty which does not take into account the number of times a token has appeared. Furthermore, if the repetition penalty is set too high in an effort to minimize this mode collapse, the sampler becomes unable to use fundamental, necessary, but frequent tokens, such as spaces or periods, resulting in poor completions. Thus, merely increasing the repetition penalty is not a viable solution either.

---

[1]See Appendix. `www.cursor.com`.

On the other hand, the frequency penalty is more adaptive. The logit penalty grows proportionally with the token count in context. However, it still fails, because it produces an (interesting) degenerate *cycle*[2] effect, where a token repeats until it incurs too high a penalty, at which point the sampler picks a new token to repeat. This is an excerpt from such a generation from an AIME question.

> **Excerpt from QwQ-32B using a frequency penalty of 0.3 and temperature of 0.**
>
> Non! Non! Non! Non! Non! Non! Non! Non! Non! Non! Non! Non! Non! Non! Non! Non! Non! Non! Non! nono! This! This! This! This! This! This! This! This! This! This! This! This! This! This! This! This! This! This! This! Third! Third! Third! Third! Third! Third! Third! Third! Third! Third! Third! Third! Third! Third! Tenth! Tenth! Tenth! Tenth! Tenth! Tenth! Tenth! Tenth! Tenth! Tenth! Tenth! Tenth! Tenth! Tenth! Tenth! Tenth! Tenth! tenth! tenth! tenth! tenth! tenth! tenth! tenth! tenth! tenth! tenth! tenth! tenth! tenth! tenth! tenth! tenth! Okay! We! We! We! We! We! We! We! We! We! We! We! We! We! We! We! We! We!

The main reason this occurs is because reasoning traces used by reasoning models such as `QwQ-32B` can become quite long, but the frequency penalty does not normalize for sequence length or account for it. Therefore, important and common tokens eventually become banned by the penalty, which degrades the completion, and eventually, results in catastrophic degeneration as seen in the excerpt.

The fundamental improvement in the LZ penalty relative to the repetition or frequency penalty is that the LZ penalty, borrowing from the sliding window matching techniques pioneered in the LZ77 (Lempel & Ziv, 1977) and LZSS (Storer & Szymanski, 1982) lossless compression algorithms, depends on the repetition of $n$-grams over a long but fixed-length sliding window. By penalizing as a function of length-normalized $n$-gram statistics as opposed to single token statistics, the penalty can be significantly more surgical in how it modulates the sampling distribution.

While there may be numerous reasonable ways to convert $n$-gram statistics into serviceable sampling penalties, we opt to base our penalty in the prediction-compression duality principle, which has various formulations, but essentially states that for every autoregressive language model, there is a dual data compression algorithm (and vice-versa). More precisely, the duality states that *logits* in a language model are equivalent, in various ways that can be formalized, to *codelengths* in a data compressor.

Following the principle, we give a quick gist of the proposed LZ penalty:

1. Simulate a universal LZ sliding window compression over the causal token sequence to compute the code: $\mathcal{C} \in \{0,1\}^*$

2. Compute the change in codelength over the alphabet for each next-token: $\Delta|\mathcal{C}| \in \mathbf{R}^{|\mathcal{A}|}$

3. Apply the change in codelengths as a penalty the model's logits (denoted $\ell$): $\ell \leftarrow \ell + \Delta|\mathcal{C}|$

Informally speaking, the interpretation of this penalty is that we are extracting the residual information in the language model after removing the information that is easily compressible by the Lempel-Ziv universal lossless data compressor. From an information-theoretic standpoint, autoregressive generation can be viewed as a sequential compression process: the model predicts each next token so as to minimize the expected codelength of the sequence under its learned distribution. In this view, both the language model and the LZ universal compressor quantify how predictable (or equivalently, how redundant) each continuation is. The LZ penalty can thus be interpreted as biasing the model's next-token distribution away from redundancies identifiable by the LZ compressor, encouraging sampling from the residual, less-compressible information.

**Limitations** While the LZ penalty substantially improves resistance to degenerate repetition, several limitations remain:

**Algorithmic complexity.** Compared to industry-standard penalties, the LZ penalty introduces additional complexity. Three hyperparameters must be set: two intrinsic to the LZ compressor (window length and buffer length) and the standard penalty strength parameter. However, similar to how the LZ hyperparameters do not need to be tuned in gzip, we find that they do not need to be tuned here.

---

[2]We refer to this as a cycle because, in principle, the model would eventually run out of new tokens and be forced to circle back to a previously used token.

**Compute overhead.** The penalty requires simulating an LZ-style compression step at each decoding step to compute codelength differentials for every token in alphabet. While the overhead is minor compared to the full forward pass of a modern large language model, and can be efficiently parallelized on GPU using operations such as PyTorch's `torch.unfold`, it is still worth consideration, especially as naive implementations can become burdensome during inference.

**Intentional Repetitions** Consider the query: *Repeat the letter "a" 100 times.* While this is a contrived query, it is a situation for which we may observe diminished performance.

**Natural Language Assumption** The LZ penalty is designed and tested for natural language modeling tasks. The LZ compression algorithm itself requires the mathematical assumptions of stationarity and ergodicity for theoretical purposes, but practically, it works quite well across almost all natural language sequences of sufficient length (at least a few hundred tokens). While variations of the LZ penalty may work for other tasks, such as multimodal, it may require a domain-specific compressor for best empirical performance.

## 2 Background and Related Works

### 2.1 Language Modeling and Sampling

Language modeling traces its origins back to Shannon's testament (Shannon, 1948), where he trained a causal language model by computing the $n$-gram frequencies over an English text corpus. Modern language models are predominantly based on the transformer architecture (Vaswani et al., 2017). Completions from transformer language models are generated by autoregressive sampling of the next-token distribution. The development of transformer language models has been accompanied by significant advancements in sampling techniques that govern text generation. Early explorations of language modeling employed top-$k$ sampling (Jozefowicz et al., 2016) to constrain the output distribution to the $k$ most probable tokens, a technique later refined in (Welleck et al., 2019), which paired it with unlikelihood training to mitigate repetition. Concurrently, temperature sampling (Bowman et al., 2016) emerged as a method to control randomness. Later, nucleus sampling was introduced as a proposed improvement over top-$k$ sampling (Holtzman et al., 2020).

These sampling strategies evolved alongside efforts to address text degeneration and repetition. GPT-2's implementation (Radford et al., 2019) implicitly utilized frequency penalties to enhance output fluency, and, concurrently, the repetition penalty (Keskar et al., 2019) was devised to prevent repetitions and encourage diversity in completions. Later, LaMDA (Thoppilan et al., 2022) applied repetition penalties to improve dialog coherence, reflecting a growing emphasis on balancing creativity and quality in LLM outputs. Together, these contributions and others eventually led to an *industry-standard* sampler which supports a temperature, a top-$k$, a top-$p$ and a frequency or repetition penalty, all of which can be used in tandem to transform the raw next token logits into a final distribution for sampling. These mechanisms are related to the theory of intrinsic motivation, which defines curiosity and creativity as progress in prediction or compression (Schmidhuber, 2010).

Recent frontier chat models have significantly improved in addressing repetition issues through advancements in training and largely no longer require a repetition penalty even for greedy decoding. Nevertheless, specialized reasoning models continue to exhibit challenges related to repetitive outputs, particularly during complex inference tasks or extended reasoning chains. While visibility is limited into closed-source reasoning models, open-source reasoning models such as DeepSeek's R1 and Qwen's QwQ both require high-temperature sampling (generally, at least 0.5 to 0.7 is recommended) in order to prevent degenerate repetitions.

**Definition 1.** *Data Sequence. A data sequence of tokens will generally be denoted by $x$ over some alphabet $\mathcal{A}$.*

We will write $x_i$ to refer to the $i$-th token in the sequence, and we will write $x_{\leq t}$ to denote the head of the sequence $(x_1, ..., x_t)$ and $x_{<t}$ to denote $(x_1, ..., x_{t-1})$. We write $x_i^t$ to denote the slice $(x_i, ..., x_t)$. We will also write $x_{>i}$ to denote the tail of a sequence.

**Definition 2.** *Causal Language Model. A causal language model, **LM** is an algorithm that maps sequences $x$ to a probability mass function (pmf) over $\mathcal{A}$.*

**Definition 3. *Cross entropy.*** *Let $H_{\mathbf{LM}}(x)$ denote the average cross-entropy loss of causal language model* **LM** *on a data sequence $x$. Recall that cross-entropy is defined:*

$$H_{\mathbf{LM}}(x) = -\frac{1}{|x|} \sum_{i=1}^{|x|} \log p_{\mathbf{LM}}(x_i \mid x_{<i}) \tag{1}$$

*where $p_{\mathbf{LM}}(x_i \mid x_{<i})$ is the probability assigned by the model to the $i$-th token given the preceding context $x_{<i}$.*

We will generally be working in the log-domain, so we write $\ell_{LM}(x)$ to denote the log-probabilities (or logits) and $p_{LM}$ to denote the corresponding pmf generated by the model given the sequence $x$: $p_{LM} = \mathbf{softmax}(\ell_{LM})$.

## 2.2 Data Compression

Data compression algorithms go back to the turn of the 20th century. In Shannon's testament (Shannon, 1948), he describes the first provably optimal compressor. Later, many entropy-optimal compressors achieved practical computational complexity assuming known data distributions. Later still, in 1977 and 1978, the first *universal* compressors were launched, LZ77 and LZ78, that could, asymptotically, achieve the entropy-rate of any stationary, ergodic data source (Lempel & Ziv, 1977; 1978; Wyner & Ziv, 1994; Morita & Kobayashi, 1993). Since then, a whole family of LZ-style compressors has emerged (Fiala & Greene, 1989; Miller & Wegman, 1985; Pavlov, 2007; Oberhumer, 1997; Yoshizaki, 1988; Storer & Szymanski, 1982; Welch, 1984).

LZ77 and LZ78 both operate on the principle of adaptively building data structures based on previously seen tokens. Imagine a scenario in which you want to train your language model from scratch while doing inference. The model updates as each new token arrives, but you also care about the model's average cross-entropy loss over the entire sequence, from start to finish, since you care about the overall compression rate. LZ algorithms are not only theoretically universal in the sense they are provably optimal for stationary ergodic data, but they are practically universal in that they generally work well on real data too, even without any prior statistical assumptions.

In this work, we will only focus on the LZ77 family, which we refer to herein as the *LZ sliding window compression algorithm*, which uses string matching from a buffer over a lookback window. This contrasts the LZ78 family, which favors tree-style dictionaries. Sliding windows are more convenient for GPUs (for example, by using PyTorch's (Paszke et al., 2019) `unfold` operation) whereas tree-based dictionary methods are more inherently sequential.

Even though all LZ sliding window algorithms work on the same basic principle of computing $n$-gram repetitions within a sliding window, they can vary in how they encode their compressed data and how they manage lookback buffers. Concretely, LZSS (Storer & Szymanski, 1982) modifies LZ77 by using a 1-bit flag to indicate whether the next chunk of data is a literal or a length-distance pair and uses literals if a length–distance pair is below a given minimum length. Since we do not actually need to encode or decode the token sequence, the details of the encoding subroutine are not particularly important for our purposes. Instead, we should focus on how many bits are required for the encodings — the *codelengths* of the resulting codes. We take LZSS as our reference compressor herein, and use the LZSS encoding scheme in the LZ penalty. When we refer to a generic LZ sliding window algorithm, we will mean the LZSS variant.

**Definition 4. *Data Compression Algorithm.*** *A data compression algorithm, or data compressor, $\mathcal{C}$ is an algorithm that injectively maps sequences over an input alphabet set $\mathcal{A}$ to binary codes $\{0, 1\}^*$.*

**Definition 5. *Single-Token Data Compressor.*** *A single-token data compressor, $\mathcal{C} : \mathcal{A} \to \{0, 1\}^*$ maps literal single-tokens to binary codes. We will assume single-token data compressors are complete prefix codes (Cover & Thomas, 2006).*

Single-token data compressors can be iteratively composed to operate over full sequences. Generally speaking, they incur a small additional overhead due to being unable to amortize over longer code blocks. Practical data compressors, however, do not encode on a single-token basis. They often operate over blocks of the full sequence.

**Definition 6.** ***Compression Rate.*** *The compression rate,* $|\bar{\mathcal{C}}|$, *for a data compressor* $\mathcal{C}$ *over sequence* $x$ *is given by* $|\bar{\mathcal{C}}|(x) = \frac{|\mathcal{C}(x)|}{|x|}$.

**LZ Sliding Window Compression Algorithm** The state of an LZ sliding window compressor is comprised of a sliding lookback window $\mathbf{w}$ and a buffer $\mathbf{b}$. LZ compressors work by encoding length-distance pairs for the buffer with respect to the lookback window. In asymptotic analysis these windows have max sizes which are allowed to grow sub-linearly in the length of the data sequence. In real implementations, they are fixed to a constant that is long enough to work practically.

**Definition 7.** *We define* `findLongestMatch` *as the following objective over input strings* $y$ *and* $z$. $d, l = \arg\max_j \left( \max_{k \le |y|} \left\{ k \mid y_{\le k} = z_j^{j+k} \right\} \right)$

**Definition 8.** ***Lempel-Ziv (LZ) Sliding Window Compressor.***

*Let* $\mathbf{w}$ *and* $\mathbf{b}$ *be sequences with* $|\mathbf{w}| > |\mathbf{b}|$.

*Let* $(D, L) \leftarrow$ `findLongestMatch`$(\mathbf{b}, \mathbf{w})$ *denote the length of the longest match to the buffer and the distance back from the end of the lookback window. Let* $\mathcal{C}'\mathcal{C}''$ *denote string-wise concatenation of codes* $\mathcal{C}'$ *and* $\mathcal{C}''$. *Then, the LZ compressor for buffer* $\mathbf{b}$ *and window* $\mathbf{w}$ *is given by:*

$$\mathcal{C}_{LZ}(\mathbf{b}|\mathbf{w}) = \begin{cases} \mathcal{C}(d, l) & \text{if } l \ge 1 \text{ and } l = |\mathbf{b}| \\ \mathcal{C}(d, l)\mathcal{C}_{LZ}(\mathbf{b}_{>l}|\mathbf{w}) & \text{if } l \ge 1 \text{ and } l < |\mathbf{b}| \\ \mathcal{C}(\mathbf{b}_1)\mathcal{C}_{LZ}(\mathbf{b}_{>1}|\mathbf{w}) & \text{if } l = 0 \end{cases}$$

**Proposition 1.** *(Storer & Szymanski, 1982) LZSS can encode a match of length* $L$ *occurring* $D$ *tokens in the past using* $|\mathcal{C}_{LZ}(L, D)| = \log L + \log D + 1$ *bits.*

On the other hand, if no match is found, we require more bits to encode a token literal.

**Proposition 2.** *(Storer & Szymanski, 1982) LZSS requires* $|\mathcal{C}_{LZ}(a)| = \log |\mathcal{A}| + 1$ *bits to encode token literals* $a \in \mathcal{A}$.

Note that the encoding scheme and algorithm state alone do not fully dictate how the LZ data compression algorithm operates in practice over a data stream. Def. 8 strictly refers to the code for a buffer sequence given a lookback window. In practice, there is some implementation-specific basic control logic used to, obviously, slide the window but also flush the buffer when codeblocks are emitted and appended to the compressed sequence. However, for the sake of simplicity, we can always *simulate* a fully populated buffer and window for a given context $x_0^t$ by setting:

$$\mathbf{b}(x) = x_{t-|\mathbf{b}|}^t \qquad \mathbf{w}(x) = x_{t-|\mathbf{b}|-1-|\mathbf{w}|}^{t-|\mathbf{b}|-1} \tag{2}$$

By always simulating a maximal buffer size, we can abstract away edge effects and the details of the implementation-specific control logic while focusing on the codelengths.

Finally, it will be helpful to define the *marginal compression* of context sequence $x$ with respect to a next token $a$.

**Definition 9.** ***Marginal Compression:*** $\Delta_a |\mathcal{C}|(x) := |\mathcal{C}(ax)| - |\mathcal{C}(x)|$ *where* $ax$ *denotes the concatenation of* $a$ *and* $x$.

We write $\Delta |\mathcal{C}|(x) \in \mathbf{R}^{|\mathcal{A}|}$ to denote a marginal compression vector indexed over the alphabet.

## 2.3 The Prediction-Compression Duality

We review the well-established duality between language modeling and data compression. The prediction-compression duality principle has numerous possible formalizations depending on the treatment of the subject, but for our purposes, we are most interested in the theme of equivalence between logits in language models

and codelengths in data compressors. We refer the reader to (Delétang et al., 2024) for a modern, in-depth treatment of prediction-compression duality.

$$\ell \sim |\mathcal{C}| \tag{3}$$

We will review one such formal treatment of the duality principle.

**Proposition 3** (Prediction–Compression Duality)**.** *Fix an alphabet $\mathcal{A}$ and a token sequence $x = x_1 \ldots x_n \in \mathcal{A}^n$.*

***Compressor $\Rightarrow$ Language-model:***

*Let* **DC** *be a single-token compressor. Define the logits of a dual language model as:*

$$\ell_{\mathbf{DC}}(x_i \mid x_{<i}) := \big|\mathcal{C}_{\mathbf{DC}}(x_i \mid x_{<i})\big| \qquad \text{(bits)}$$

*by the codelength it assigns to $x_i$ conditioned on the history $x_{<i}$.*

*Define the causal probability assignment $p_{\mathbf{DC}} = \mathbf{softmax}(\ell_{\mathbf{DC}})$ as usual.*

*Then the compression rate of* **DC** *equals the per-token cross-entropy of the induced language model:*

$$\big|\bar{\mathcal{C}}_{\mathbf{DC}}\big|(x) = \frac{1}{n}\sum_{i=1}^{n} \ell_{\mathbf{DC}}(x_i \mid x_{<i}) = -\frac{1}{n}\sum_{i=1}^{n}\log p_{\mathbf{DC}}(x_i \mid x_{<i}) = H_{\mathbf{DC}}(x) \ \text{bits/token.}$$

***Language-model $\Rightarrow$ Compressor:***

*Let* **LM** *be any causal language model that outputs $p_{\mathbf{LM}}(\cdot \mid x_{<i})$.*

*Then, the Arithmetic coding construction ((Cover & Thomas, 2006; Witten et al., 1987)) produces a sequential prefix-free compressor $\mathcal{C}_{\mathbf{LM}}$ satisfying, for every $x \in^n$,*

$$\big|\bar{\mathcal{C}}_{\mathbf{LM}}\big|(x) = \frac{1}{n}\sum_{i=1}^{n}\big|\mathcal{C}_{\mathbf{LM}}(x_i \mid x_{<i})\big| \leq -\frac{1}{n}\sum_{i=1}^{n}\log_2 p_{\mathbf{LM}}(x_i \mid x_{<i}) + \frac{2}{n} = H_{\mathbf{LM}}(x) + O(1/n) \ \text{bits/token.}$$

*Hence, up to an asymptotically negligible $O(\frac{1}{n})$ redundancy,*

$$\big|\bar{\mathcal{C}}_{\mathbf{DC}}\big|(x) = H_{\mathbf{DC}}(x), \qquad \big|\bar{\mathcal{C}}_{\mathbf{LM}}\big|(x) = H_{\mathbf{LM}}(x).$$

Given a language model, we also have a data compressor that compresses as well as the language model predicts, and given a data compressor, we have a language model that predicts as well as that data compressor can compress. The Arithmetic code (and other codes such as the Huffman code (Huffman, 1952; Cover & Thomas, 2006)), employ the prediction-compression duality to assign codelengths based on log-probabilities.

The situation is more complex for constructing causal language models from online data compressors such as LZ sliding window algorithms. This is because causal language models must be able to generate a valid next-token pmf at every step whereas data compressors often buffer tokens together into a single code. Practically, this means data compressors do not necessarily produce a codelength for every next-token. We address this issue by *simulating* a full buffer and lookback window at each next-token. Similar ideas have been explored in (Ryabko, 2007).

## 3 LZ Penalty

The core essence of the LZ penalty is to use the prediction-compression duality to construct a compressor's dual language model (in this case, LZSS)[3]. We can then apply the following logit update to the language model we wish to penalize, for some penalty strength $0 \leq \alpha$:

---
[3]Refer to Fig. 1 for an architecture diagram of the LZ penalty.

$$\ell_{LM} \leftarrow \ell_{LM} + \alpha\Delta|\mathcal{C}_{LZ}| \tag{4}$$

where $\Delta|\mathcal{C}_{LZ}|$ is the marginal compression under a simulated LZ sliding window compressor due to each potential next-token. Note that adding a redundant token can actually *shorten* the full codelength under the LZ compressor, which results in a *negative* marginal codelength to penalize overly redundant tokens.

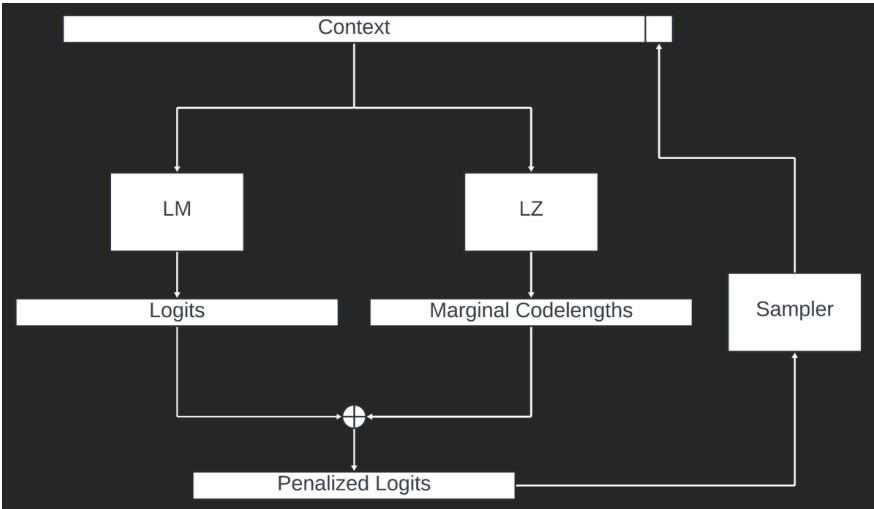

Figure 1: An architecture diagram detailing the flow of an autoregressive sampling loop using the LZ penalty.

Let $x$ denote the current context. We simulate the LZ sliding window $\mathbf{w}(x)$ and buffer $\mathbf{b}(x)$ as in (2). We can then compute the incremental change in codelength due to each possible next-token $a \in \mathcal{A}$ relative to the simulated buffer and window. We can then compute the *simulated* marginal compression for LZSS for all $a \in \mathcal{A}$:

$$\Delta|\mathcal{C}_{LZ}|(x) := \Delta|\mathcal{C}_{LZ}|\,(\mathbf{b}(x)|\mathbf{w}(x)) = |\mathcal{C}_{LZ}(a\mathbf{b}(x)|\mathbf{w}(x))| - |\mathcal{C}_{LZ}(\mathbf{b}(x)|\mathbf{w}(x))| \tag{5}$$

Since we are operating in the log-domain under softmax affine invariance:

$$\Delta|\mathcal{C}_{LZ}|(x) \propto |\mathcal{C}_{LZ}(a\mathbf{b}(x)|\mathbf{w}(x))| \tag{6}$$

Going forward, we omit explicit dependence on $x$ and $LZ$ when it is obvious.

Let $d, l \leftarrow \texttt{findLongestMatch}(\mathbf{b}(x), \mathbf{w}(x))$ and $\delta, \lambda \leftarrow \texttt{findLongestMatch}(a\mathbf{b}, \mathbf{w})$.

If $l = 0$, we know the virtual next-token $a$ comes after a literal in the encoding. This implies that:

$$\lambda(a) = \begin{cases} 1 & \text{if } a \in \mathbf{w}(x) \text{ and } l = 0 \\ 0 & \text{if } a \notin \mathbf{w}(x) \text{ and } l = 0 \end{cases}$$

with $\delta$ giving the distance of the match (if present).

If $l \geq 1$, then the virtual next token might extend a match. In the case that it does so, then $\delta = d - 1$ and $\lambda = l + 1$, because the match location shifts one spot to the right and the length increases by one. If it does not extend the match, then $\lambda \leq 1$.

We proceed with a case-by-case calculation of $|\mathcal{C}_{LZ}(a\mathbf{b}|\mathbf{w})|$. Recall we are working in the log-domain, and that because $l, d$ are independent of $a$, due to softmax affine invariance, we can ignore terms that only depend on $l, d$ but are constant with respect to the choice of $a$.

**Case I:** ($l = 0$)

$$\mathcal{C}_\mathrm{I}(a\mathbf{b}|\mathbf{w}) = \mathcal{C}(a|\mathbf{w})\mathcal{C}(\mathbf{b}|\mathbf{w}) \implies |\mathcal{C}_\mathrm{I}(a\mathbf{b}|\mathbf{w})| = |\mathcal{C}(a|\mathbf{w})| + |\mathcal{C}(\mathbf{b}|\mathbf{w})| \propto |\mathcal{C}(a|\mathbf{w})| \tag{7}$$

Furthermore, as discussed above:

$$\mathcal{C}_\mathrm{I} = \mathcal{C}(a|\mathbf{w}) = \begin{cases} \mathcal{C}(1,\delta) & \text{if } \lambda = 1 \\ \mathcal{C}(a) & \text{if } \lambda = 0 \end{cases} \tag{8}$$

Where $\mathcal{C}(1,\delta)$ encodes a singleton match at distance $\delta$ and $\mathcal{C}(a)$ encodes $a$ as a literal. This gives us our first case: $\mathcal{C}_\mathrm{I} \propto |\mathcal{C}(a|\mathbf{w})|$.

Recalling Prop. 1 and 2 and removing constants due to softmax affine invariance, we obtain simple expressions:

$$|\mathcal{C}_\mathrm{I}| = \begin{cases} \log \delta & \text{if } \lambda = 1 \\ \log |\mathcal{A}| & \text{if } \lambda = 0 \end{cases} \tag{9}$$

**Case II:** ($l \geq 1$)

$$\mathcal{C}_\mathrm{II}(a\mathbf{b}|\mathbf{w}) = \mathcal{C}(a\mathbf{b}_{\leq l}|\mathbf{w})\mathcal{C}(\mathbf{b}_{>l}|\mathbf{w}) \implies |\mathcal{C}_\mathrm{II}| = |\mathcal{C}(a\mathbf{b}_{\leq l}|\mathbf{w})| + |\mathcal{C}(\mathbf{b}_{>l}|\mathbf{w})| \propto |\mathcal{C}(a\mathbf{b}_{\leq l}|\mathbf{w})| \tag{10}$$

Furthermore:

$$\mathcal{C}_\mathrm{II} = \mathcal{C}(a\mathbf{b}_{\leq l}|\mathbf{w}) = \begin{cases} \mathcal{C}(a|\mathbf{w})\mathcal{C}(\mathbf{b}_{\leq l}|\mathbf{w}) = \mathcal{C}(a|\mathbf{w})\mathcal{C}(l,d) & \text{if } \lambda \leq 1 \\ \mathcal{C}(a\mathbf{b}_{\leq l}|\mathbf{w}) = \mathcal{C}(\lambda, \delta) & \text{if } \lambda = l+1 \end{cases} \tag{11}$$

Recall that, as discussed above $\lambda \leq 1$ if and only if $a$ does not extend the match of length $l$. If $a$ does extend the match, then $\lambda = l + 1$. Reusing 8, we can further simplify:

$$\mathcal{C}_\mathrm{II} = \begin{cases} \mathcal{C}(a)\mathcal{C}(l,d) & \text{if } \lambda = 0 \\ \mathcal{C}(1,\delta)\mathcal{C}(l,d) & \text{if } \lambda = 1 \\ \mathcal{C}(\lambda,\delta) & \text{if } \lambda = l+1 \end{cases} \tag{12}$$

Again reusing Prop. 1 and 2:

$$|\mathcal{C}_\mathrm{II}| = \begin{cases} |\mathcal{C}(a)\mathcal{C}(l,d)| = |\mathcal{C}(a)| + |\mathcal{C}(l,d)| = \log |\mathcal{A}| + \log(ld) + 1 & \text{if } \lambda = 0 \\ |\mathcal{C}(1,\delta)| + |\mathcal{C}(l,d)| = \log(\delta) + \log(ld) + 1 & \text{if } \lambda = 1 \\ |\mathcal{C}(\lambda,\delta)| = \log(\lambda\delta) & \text{if } \lambda = l+1 \end{cases} \tag{13}$$

It is expedient and permissible (due to affine invariance) to subtract the $\log(ld) + 1$ term.

$$|\mathcal{C}_\mathrm{II}| = \begin{cases} \log |\mathcal{A}| & \text{if } \lambda = 0 \\ \log(\delta) & \text{if } \lambda = 1 \\ \log(1 - \frac{d-l+1}{ld}) - 1 & \text{if } \lambda = l+1 \end{cases} \tag{14}$$

where $\log(1 - \frac{d-l+1}{ld}) = \log(\lambda\delta) - \log(ld)$ follow from $\lambda = l+1$ and $\delta = d - 1$.

**LZ Penalty Formula** Combining cases I and II above yields a complete formula for the LZ penalty adjustment:

$$\Delta|\mathcal{C}_{LZ}| = \begin{cases} \log|\mathcal{A}| & \text{if } \lambda = 0 \\ \log(\delta) & \text{if } \lambda = 1 \\ \log(1 - \frac{d-l+1}{ld}) - 1 & \text{if } \lambda = l + 1 \end{cases} \tag{15}$$

Assuming $|\mathcal{A}| > |\mathbf{w}|$, then this provides a dynamic range of $[\log(2/|\mathbf{b}|) - 1, \log|\mathcal{A}|]$. For an alphabet of size $128k$, a lookback window of size 512, and a buffer of size 32, using binary logarithms, this yields an adjustment range from $-5$ to $+17$, with a $-5$ adjustment going to a token that would complete an immediate repetition of length 32 and a $+17$ going to a token that does not appear in the previous 512 tokens.

## 4 Results

We perform an empirical study of how the LZ penalty affects repetition and capability in reasoning benchmarks and a performance study of the `SGLang` reference implementation.

### 4.1 Repetition and Capability Benchmarks

We apply the LZ penalty to `QwQ-32B`[4] and `R1-Distill-14B`[5]. We run GPQA and AIME benchmarks (averaging scores and computing std. dev. over 5 runs). We set a max token limit of $24k$. We fixed the `top-p` to 0.95 and the `top-k` to 40 for all runs.[6] For all runs also using the LZ penalty, we fix the penalty strength $\alpha$ to 0.15, the window size to 512 and the buffer size to 32. We found that this configuration of hyperparameters seemed to work well across both models and both datasets with minimal tuning required.[7] We detect degenerate repetitions via dual verification of a `GPT-4o` based judge and a naive search for exact repetitions.[8]

**Baselines** We compare the LZ penalty against two industry-standard penalties: the repetition penalty and the frequency penalty. In both cases, we finely sweep small values up until getting to large values. For the results of the full sweep of penalty values, refer to the Appendix.

**Discussion** Based on Fig. 2, we observe that neither penalty is a reliable solution. The frequency penalty fails dramatically even for low values. We suspect that this is because of the length of the generations. Reasoning models produce reasoning traces that can be several thousand tokens long, which simply overwhelms the frequency penalty on common but essential tokens. The repetition penalty works significantly better than the frequency penalty and does seem to provide some modest relief. However, it is far from a complete solution, with low temperature degenerate repetition rates up to about $\sim 4\%$ depending on model and task domain. This would be disqualifying for any kind of serious application. On the other hand, the LZ penalty achieves effectively zero degenerate repetitions without affecting top-line benchmark scores. The LZ penalty works because it adaptively penalizes based on both the length of the match as well as how far back the match occurs. LZ penalty's strength increases quickly in match length and attenuates gradually with distance. Neither the repetition penalty nor the frequency penalty can forget tokens, whereas the LZ penalty quickly and then gradually weakens as the token becomes less recent, until it moves beyond the lookback window altogether.

Figure 2: Line charts showing the accuracy and repetition percentage for a baseline (repetition penalty of 1, frequency penalty of 0), the LZ penalty, the repetition penalty, and the frequency penalty. Accuracy error bars indicate the empirical std. dev. over 5 runs. We feature the best performing choice of repetition penalty and frequency penalty strengths.

| Model Size | Med. Latency (ms) | Med. Throughput (tok/s) | Slowdown (%) |
|---|---|---|---|
| 1.5B | 4.43 | 14449.88 | – |
| 1.5B + LZ | 4.45 | 14370.98 | 0.55 |
| 7B | 7.96 | 4020.06 | – |
| 7B + LZ | 7.97 | 4014.29 | 0.14 |
| 32B | 26.71 | 299.55 | – |
| 32B + LZ | 26.71 | 299.47 | 0.03 |

Table 1: Median latency, throughput, and LZ penalty's throughput slowdown for Qwen-2.5 architecture using `SGLang`'s default benchmarking script. Context length: 1024, generation length: 64. Batch sizes: 64 (1.5B), 32 (7B), 8 (32B).

## 4.2 Latency and Throughput Benchmark

Although our `SGLang` reference implementation is not fully optimized, it is vectorized and batched. We run `SGLang`'s built-in benchmark script on an 8×H100 node and compare the effect of adding non-zero LZ penalty. While the LZ penalty adds an ultimately negligible amount of computation, it still is significantly more than, say, the repetition penalty, so it is worthwhile to confirm that we can maintain inference performance.

We see that for larger models, the LZ penalty's overhead is increasingly negligible. Even for models as small as 1.5B, the penalty overhead is a tolerable 0.55% throughput slowdown. For latency, the overhead is more trivial and is not even measurable at the 32B size.

## 5 Conclusion

We presented the Lempel-Ziv (LZ) penalty, an information-theoretic decoding strategy that suppresses degenerate repetitions in autoregressive language models by leveraging the prediction–compression duality. Unlike frequency and repetition penalties, the LZ penalty adaptively accounts for both match length and recency, enabling reasoning models to decode *deterministically* without loss of capability. Empirical studies show that the LZ penalty eliminates degenerate loops while preserving reasoning benchmark accuracy, with negligible computational overhead. These findings suggest that compression-informed penalties offer a principled and practical path toward more reliable language model decoding.

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
