# OpenReview forum: "LZ Penalty: An information-theoretic repetition penalty for autoregressive language models."
_TMLR — Accepted by TMLR_

### Review · Reviewer_aPYm · 2025-08-20

**Summary Of Contributions:**

The paper proposes a decoding "penalty" for LLMs that is based on the Lempel-Ziv77 compression algorithm. Intuitively, the method increases logits with a large code length more and increases less (or decreseases) logits with a small code length. In contrast to previous methods like frequency or repitition penalty, the proposed LZ penalty considers length-normalized n-gram statistics as opposed to single token statistics. This way, the penalty is more "surgical" in how it modulates the sampling distribution and thereby prevents degenerate cycling effects common with frequency penalty, while being more nuanced than the binary repitition penalty.

**Strengths**

Overall, I think it is a great paper. I enjoyed reading it (apart from the rendering issues).

1. The proposed LZ penalty is plausible and works well, as clearly shown in the experiments.

2. The proposed LZ penalty seems easy to integrate.

3. Overall, the paper is easy to read (though some things could be written more clearly, see *requested changes*).

**Weaknesses**

1. A proper related work section is missing (there is only a background section) and generally, there are few references. I am not an expert at LLM decoding algorithms, but I would be suprised if not more similar ideas (besides [1], which the authors cite) exist. If indeed there is little, one could broaden the scope. For instance, I feel loosely reminded of [2] who incentivize an "artist/scientist" agent to generate data that is not easily compressable by some compression method. I am not saying that [2] is the paper that must be discussed: It is likely that more closely related works than [2] exist by now. Anyway, a related work section of some kind should be present so the reader can better categorize the contribution within the existing literature.

2. A limitations section is also missing. As a reader, I need to guess what the shortcomings of the proposed method are and I believe that the authors need to state and discuss this more clearly in such a dedicated section (TMLR is not like the toxic ML conferences, so there should be no incentive to be secretive about limitations). I believe the main shortcomings are (please correct me if I am wrong): (1) Three tuning parameters instead of just a single one for the baselines (repition and frequency penalty) (2) Additional runtime due to the $\texttt{findLongestMatch}$ function, which seems quadratic in complexity (experiments for this point exist, which is good, but no critical discussion). Please add such a section to discuss these and potentially further limitations that I did not catch.

3. A conclusion is also missing. I believe this is important.


[1] Boris Ryabko. "Compression-based methods for nonparametric density estimation, on-line prediction, regression and classification for time series" (2007).

[2] Schmidhuber, Jürgen. "Formal theory of creativity, fun, and intrinsic motivation (1990–2010)." IEEE transactions on autonomous mental development 2.3 (2010): 230-247.

**Additional Comments:**

I decided to write a review for the paper in spite of the rendering issues, which was quite annoying. To show respect towards the reviewers, I would kindly ask the authors to check for such issues more carefully before uploading a paper next time.

**Audience:**

No

**Audience Explanation:**

The paper is not rendered correctly. I would normally not read a paper with such rendering issues. Therefore, I deem it not interesting.

**Claims And Evidence:**

Yes

**Claims Explanation:**

A descent experimental evaluation is present.

**Requested Changes:**

I will proceed to list requested changes one-by-one. Points that are critical for meeting the acceptance criteria are highlighted by *(critical)*.

**Overall**

- *(critical)* The rendering must be fixed. In the current state, the paper must not be accepted.

- *(critical)* A discussion/limitations section is missing.

- *(critical)* A conclusion section is missing.

**Title**

This is highly subjective, but I do not like acronyms in the title. Would it be possible to just write out Lempel-Ziv penalty in the title? I would do that as long as there is not good reason against doing so.

**Abstract**

- *"Both the industry-standard frequency penalty and repetition penalty are
ineffective, incurring degenerate repetition rates of up to 4% or more."* If you state how bad existing methods are, I would expect that you write how much better your method is in comparison. Otherwise, the statement is meaningless. Please highly consider adding that info in the abstract.

**1 Introduction**

- *"In recent months, there has been an advent in reasoning models."* Please consider adding some references to support this claim.

- *"Reasoning models do not run well at low temperatures because the sampling distribution
mode collapses into degenerate repetitions."* This sentence is out of context to me. The previous sentence was about lack of determinism around the reasoning trace and now we talk about problems at low temperatures.

- *"Furthermore, if the repetition penalty is set too high"* So why not just tune the penalty then? Please consider adding an explanation for why that is difficult or cannot be done.

- *"On the other hand, the frequency penalty is more adaptive"* Why *"on the other hand"*? Where is the *"on the one hand"*? Consider instead writing *"In constrat to the repition penalty, the frequency penalty is more adaptive"*

- *"Ultimately, this still fails. Instead, it produces an interesting degenerate"* This is an odd formulation. Why not just write *"However, it still fails, because it produces an (interesting) degenerate ..."*?

- *"This is a real excerpt"* Please consider dropping *"real"*... I hope that all of the results presented in this paper are real.

- *"Excerpt from QwQ-32B using a frequency penalty of 0.3 and temperature of 0"* From my understanding, this cannot be. If the temperature is truly $0$, then you divide by $0$, no? Please consider clarifying.

- *"Following the principle, we give a quick gist of the proposed LZ penalty:"* I like the simplified three-step presentation. However, the notation is not clear. For instance, what is $l$? A logit? I would suggest either explaining the symbols or omitting them (the latter is ok, given that this is the introduction).

**2.1 Language Modeling and Sampling**

- *where he trained a casual language mode*. Typo: "casual" -> "causal".

- *Definition 1. Data Sequence.* This is a subjective comment, but I find all of these definitions a bit of an overkill. This paper is not very mathematical, so I do not see why the notation needs to be introduced like this.

**2.3 The Prediction-Compression Duality**

- *Proposition 3* I believe that an additive constant is missing in the final equation, because of the normalization. I.e., I think it should be $ \frac{1}{n} \sum_{i=1}^n \ell_{DC} (x_i | x_{<i})  =  -\frac{1}{n} \sum_{i=1}^n \text{log}  p_{DC} (x_i | x_{<i}) + const.$.

- *"Constructing casual language models"* Typo: "casual" -> "causal".

**3 LZ Penalty**

- *"duality to construct a an LZ"* Typo: Remove "an".

- *equation (3)* The equation makes sense, but I believe the terminology is incorrect. Let me clarify: If a token has not been sampled often yet, then it has a large code length and its probability of being sampled increases. Likewise, if a token has already been used many times, its codelength is smaller and its probability of being sampled will be smaller compared to "rarer" tokens. Is that correct? If yes, then *penalty* does not seem like the right word, because penalization would imply that a large penalty should make the token less likely, no? Anyway, using such an example like I just did would greatly help understanding what is going on.

- *this yields on adjustment range from −5 to +17, with a −5 adjustment going to a token that would complete an immediate repetition of length 32 and a +17 going to a token that does not appear in the previous 512 tokens.* This seems to confirm my belief of how this works. However, as mentioned in the comment above, I believe the term *penalty* could be missleading. I would suggest replacing this term by *adjustment* or *offset* or something like that.

**Results**

- *(critical)* *"For all runs also using the LZ penalty, we fix the penalty
strength $\alpha$ to 0.15, the window size to 512 and the buffer size to 32"* In the introduction, the authors were mentioning that it is difficult to tune the repition penalty for the baseline. However, it seems that the proposed method has three parameters that all need to be tuned (not just a single one). Something like this must be critically discussed.

---

> ### Comment · Reviewer_aPYm · 2025-09-01
>
> I thank the authors for fixing the rendering and implementing my suggestions. I would like to make a few comments:
>
> 1. This is minor, but would it be possible to turn the limitations section into a proper section and not just a paragraph? Currently, the limitations section is a paragraph, but the concrete limitations are also paragraphs. This may be confusing to the reader, since the concrete limitations should be on a sub-ordinate level to the limitations section.
>
> 2. I believe that the authors mainly renamed the "Background" section to "Background & Related Work" and only added one more reference (which is the one I provided). As already mentioned, I am not an expert in LLM decoding algorithms. I am willing to trust the authors that no more closely related works exist. Yet, I must admit that I am surprised. I would be happy if other reviewers who know more could check whether this is actually the case.
>
> 3. Regarding the revision, it would be nice if the authors could in the future (a) respond to reviews in the designated field and not just in the general *Changes Since Last Submission:* field; and (b) highlight the changes in the manuscript in some color. It is very hard to see what the authors have changed in the new version. I would like to point out that TMLR reviewers are not being paid and therefore I do not think it is too much to expect from the authors to properly guide reviewers in the reviewing process.

---

> > ### Comment · Action_Editor_CQZG · 2025-09-09
> > **OpenReview has inbuilt diff feature**
> >
> > Dear aPYm,
> > thanks for your timely review and engagement with the authors. Just wanted to point out that OpenReview also has a built in feature to compare PDF revisions. You can find it by clicking on 'Revisions' just below the title and author list. It's not as nice as authors highlighting changes directly (especially when figures / tables are moved between pages), but better than having no comparison.
> >
> > We are still waiting for the other two reviews, but in the meantime the authors can respond to your additional comments.

---

> > > ### Comment · Reviewer_aPYm · 2025-09-09
> > >
> > > Thank you, I did not know about this revision feature. It helps a lot. Having it highlighted would still be better, as you write. For the future, I would still highly recommend this to the authors.

---

### Review · Reviewer_cxDk · 2025-10-06

**Summary Of Contributions:**

This paper demonstrate the LZ penalty enables open-source reasoning models to operate with greedy decoding without loss of capability and without instances of degenerate repetition.

**Audience:**

Yes

**Audience Explanation:**

This is a very typical regularization approach so if we can have a better understanding of this method, we might help the model learn the target distribution more efficiently.

**Claims And Evidence:**

No

**Claims Explanation:**

It is not clear what's the assumption behind this paper. LZ77 and corresponding compression algorithms does not seem to be a natural correspondnece to real world data. (As real world data might have a lot of discrepancy across data distributions). It is necessary to discuss when and how this LZ77 will help the performance of the model .

**Requested Changes:**

Try to provide examples of when and how LZ77 type regularization method do not help. Analyze when to use this method how to make the performance maximized. If a clear theoretical justification is hard to achieve, feel free to provide folklore statement.

---

> ### Comment · Reviewer_aPYm · 2025-10-07
>
> I agree that it would be nice to see failure modes of the method and some intuition for why they occur. This will be useful for both practitioners who want to know when not to use the method and researchers who want to identify new problems to tackle.

---

> ### Author Response · Authors · 2025-10-31
> **Response from Authors**
>
> We appreciate the reviewer’s time and thoughtful feedback. We believe some of the confusion arises from a misinterpretation of the role of the LZ77 compressor in our method. To clarify, our use of “LZ77” is not intended as a literal modeling assumption about the distribution of natural language data, but rather as a universal reference compressor within the information-theoretic framework. The LZ penalty is not trained to emulate real-world data distributions; instead, it leverages the prediction–compression duality to provide a principled and dynamic measure of redundancy during decoding.
>
> Specifically:
>
> LZ77 is universal in the sense that, for any stationary ergodic source, it asymptotically achieves the optimal compression rate without prior assumptions on the true distribution. We therefore use it as a canonical, assumption-free estimator of compressibility.
>
> The LZ penalty applies this compressibility signal online during decoding, biasing the model’s next-token distribution away from redundancies identifiable by the universal compressor. The mechanism does not assume that real-world data literally follows the LZ model.
>
> Empirically, we find that this universal bias aligns well with the structure of reasoning sequences and reduces degenerate repetitions across diverse tasks, without loss of capability.
>
> Regarding the reviewer’s suggestion to discuss failure cases: we agree that exploring when the LZ penalty may be less effective is valuable. We have added a brief discussion in the Limitations section noting that tasks requiring intentional repetition (e.g., generating repeated characters or patterned text) are expected to see diminished performance, since the penalty explicitly discourages locally compressible sequences.
>
> We thank the reviewer for prompting this clarification, and have added language in the revised version to make the distinction between universal compression as a theoretical construct and empirical data distributions explicit.

---

### Review · Reviewer_pfie · 2025-10-06

**Summary Of Contributions:**

This paper introduces a technique to mitigate repetition issues in large language model (LLM) outputs through the lens of code compression. Experiments on various mid-sized LLMs demonstrate that the proposed method significantly reduces repetition compared to the baseline while maintaining comparable accuracy. The main weakness is that the contribution seems a little marginal compared to repetition penality.

**Additional Comments:**

Typos and cross-references. There are several typographical errors (e.g., “ir still fails” → “it still fails”) and incorrect figure references. In particular, the sentence in Section 4.1 should cite Figure 2, not Figure 4.

**Audience:**

Yes

**Audience Explanation:**

The repetition issue in large language models (LLMs) is a universally important problem, particularly for smaller models. This paper investigates this important topic and provides valuable new algorithm into mitigating repetition.

**Claims And Evidence:**

Yes

**Claims Explanation:**

The experiments, particularly those shown in Figure 2, demonstrate that the proposed algorithm serves as a strong alternative to existing baselines such as repetition penalty and frequency penalty.

**Requested Changes:**

The compression perspective is not clearly presented in the introduction, which defers the reader’s understanding to much later sections. I suggest improving the intuitive explanation — for instance, it would be helpful to clarify what is meant by “compression of the generation.”

Secondly, while the reduction in repetition shown in Figure 2 is noticeable, the improvement over the repetition penalty baseline appears relatively marginal. Moreover, the accuracy on R1-DISTILL-14B-AIME is even lower than that of the repetition penalty method. I recommend enhancing the proposed algorithm to better demonstrate its practical effectiveness and ability to outperform existing repetition-control baselines.

Furthermore, how does your method perform in smaller models where the repitition errors are more phenomenal?

---

> ### Author Response · Authors · 2025-10-31
> **Response From Authors**
>
> We sincerely appreciate your time and thank you for your thoughtful review.
>
> In response, we have:
>
> (1) Corrected typographic errors.
> We identified and corrected nearly a dozen typographic mistakes, using AI-based proofreading to ensure thorough removal of typos in every section.
>
> (2) Clarified the introduction.
> Regarding your question about the introduction, we added the following passage to more clearly establish the link between compression and generation:
>
> Informally speaking, the interpretation of this penalty is that we are extracting the residual information in the language model after removing the information that is easily compressible by the Lempel–Ziv universal lossless data compressor. From an information-theoretic standpoint, autoregressive generation can be viewed as a sequential compression process: the model predicts each next token so as to minimize the expected codelength of the sequence under its learned distribution. In this view, both the language model and the LZ universal compressor quantify how predictable (or equivalently, how redundant) each continuation is. The LZ penalty can thus be interpreted as biasing the model’s next-token distribution away from redundancies identifiable by the LZ compressor, encouraging sampling from the residual, less-compressible information.
>
> (3) Addressed repetition performance.
> Regarding the comment on repetition performance being marginal: while we agree that degenerate repetitions are uncommon, as we show, they can still occur up to 4% of the time even for the best existing repetition penalty (depending on the dataset and model). The purpose of this work is to show that the LZ penalty provides, as closely as possible given the inherently stochastic nature of LLMs, a guarantee against degenerate repetitions. We believe this constitutes a major improvement for two reasons:
>
> (3a) While 4% may sound small, degenerate repetitions are an extremely poor user experience (see Fig. 3 in A.1) that still occur in real-world settings.
> (3b) In terms of relative decrease, the improvement is actually very large—at least a >40× reduction, if not “infinite.”
>
> Taking a small but painful error case and reducing it to negligible frequency remains an important contribution.
>
> (4) Clarified the R1-Distill-14B results.
> We believe that Fig. 2 shows no significant degradation in AIME benchmark performance. Measurements are within error bars of each other for virtually every temperature, and moreover, the “better” performing group fluctuates slightly as we scan temperatures—indicative of random noise rather than systematic bias.
>
> (5) Explained model size choices.
> Regarding smaller models: the choice of two model sizes, 14B and 32B, was made to reflect a medium-sized regime balancing the need to expose degenerate repetitions with maintaining model capability and downstream usefulness. Most “small” model offerings from leading APIs are in this range or even larger. Finally, we note that even frontier models—presumably in the hundreds of billions to trillions of parameters—still exhibit degenerate repetitions (see A.1).

---

### Author Response · Authors · 2025-12-08
**[Reviewer action needed] Submit final recommendation**

Dear Reviewers,

Please let us know if there is anything else we can do or further clarify any questions you might have.

If you are otherwise satisfied, we would greatly appreciate your final review submission.

Thank you so much for taking the time to review our work and for your valuable feedback!

Best,
Authors

---

### Decision · Action_Editor_CQZG · 2025-12-07

**Recommendation:** Accept with minor revision

**Additional Comments:**

Below is a list of items for a minor revision of the manuscript:

1. Please do another pass for formatting and typesetting issues (I have listed some concrete ones below, but also please check if some paragrahps should be merged or split; for example the paragraph that starts with "Concretely, LZSS" looks like it should be part of the previous paragraph), and fixing typos.
1. The intro uses footnotes to include links to some common LLMs etc. Consider using references instead (most LLMs have a standard reference, typically on arxiv, but one can also add blog posts etc. to a bibliography). This is a suggestion, not strictly necessary.
 1. Section 2.1 and 2.2 mixes related work and background. This is fine, though I personally would prefer separating all the related work discussion from introducing the mathematical/formal background, e.g. doing related work first and then the background.
 1. The term ‘vocabulary V’ is used throughout the paper, but unless I am mistaken what the authors mean is the ‘alphabet A’. The alphabet is the set of all atomic tokens, whereas typically a vocabulary refers to a set of (short) sequences of tokens.
 1. Definition 3: please write down the expression for the log-loss (cross-entropy loss).
 1. Sec. 2.3. Please add and briefly discuss the following reference: “Language modeling is compression, Deletang et al. 2023” - it makes exactly the point in 2.3. in greater detail.
 1. Sec. 4.1. Please give more details about the GPT-4o based judge for repetition detection (e.g. in the appendix).
 1. Fig. 1 seems to have no reference in the main text.
 1. Typo "Intential Repetitions" -> "Intentional Repetitions".
 1. Add a detailed discussion of the ergodicity assumption to the limitations section (or another part of the paper) and a statement that provides clarity about when LZ-based compressibility is theoretically appropriate and when it is not and how that relates to LLM use-cases.

**Audience:**

Yes

**Audience Explanation:**

The topic of improving sampling from foundation models is interesting to TMLR's audience. The paper makes an interesting proposal that works well empirically (in the particular cases tested), which has good potential to inspire follow-up research, improvements, and extensions by the community.

**Claims And Evidence:**

Yes

**Claims Explanation:**

The paper proposes a novel technique for sampling from foundation models under low temperature that avoids recurring output patterns---to be precise: sequences that are highly compressible under LZ77 are discouraged via an additive logit penalty term. The idea is interesting and has theoretical merit, though in practice it might take LZ77 "a while" before strongly penalizing repetitions, and of course, since the method only works on token-level, it is blind to semantic repetition expressed via different token sequences. The main aim of the method is to avoid degeneration in thinking traces in modern LLMs, and the limited empirical evaluations shown in the paper provide evidence that the method works well in this setting without degrading sample quality.

All reviewers agree that the paper's claims are supported by convincing evidence, and I agree that the paper passes the bar. I think the manuscript needs another pass for formatting and other minor issues, but overall the work is ready to be shared more widely. I therefore suggest some items for a minor revision below, and accepting the paper with these revisions.

---

> ### Author Response · Authors · 2025-12-17
> **Thank you from the authors**
>
> We'd like to thank the editor and reviewers for the feedback and helpful suggestions. We are excited to be able to share this work!
>
> We will promptly proceed with these revisions and share an updated camera-ready manuscript soon.

---

> ### Author Response · Authors · 2026-01-23
> **Camera Ready Response**
>
> Thank you for the valuable feedback. We've addressed your points as follows:
>
> 1. We have done another pass and fixed typesetting issues and typos
> 2. Citations added
> 3. We have re-ordered the content in each section to be background first then formalism.
> 4. Replaced vocabulary V with alphabet A
> 5. Added
> 6. Yes, this is an excellent reference. Added and thank you for pointing it out.
> 7. Added
> 8. Added reference
> 9. Fixed
> 10. Added